# Genome-Wide Identification and Expression Analysis of the Metacaspase Gene Family in *Gossypium* Species

**DOI:** 10.3390/genes10070527

**Published:** 2019-07-12

**Authors:** Senmiao Fan, Aiying Liu, Zhen Zhang, Xianyan Zou, Xiao Jiang, Jinyong Huang, Liqiang Fan, Zhibin Zhang, Xiaoying Deng, Qun Ge, Wankui Gong, Junwen Li, Juwu Gong, Yuzhen Shi, Kang Lei, Shuya Zhang, Tingting Jia, Lipeng Zhang, Youlu Yuan, Haihong Shang

**Affiliations:** 1State Key Laboratory of Cotton Biology, Key Laboratory of Biological and Genetic Breeding of Cotton, The Ministry of Agriculture, Institute of Cotton Research, Chinese Academy of Agricultural Sciences, Anyang 455000, China; 2School of Agricultural Sciences, Zhengzhou University, Zhengzhou 450000, China

**Keywords:** cotton, metacaspase, evolution, expression patterns

## Abstract

Metacaspases (MCs) are cysteine proteases that are important for programmed cell death (PCD) in plants. In this study, we identified 89 *MC* genes in the genomes of four *Gossypium* species (*Gossypium raimondii*, *Gossypium barbadense*, *Gossypium hirsutum*, and *Gossypium arboreum*), and classified them as type-I or type-II genes. All of the type-I and type-II *MC* genes contain a sequence encoding the peptidase C14 domain. During developmentally regulated PCD, type-II *MC* genes may play an important role related to fiber elongation, while type-I genes may affect the thickening of the secondary wall. Additionally, 13 genes were observed to be differentially expressed between two cotton lines with differing fiber strengths, and four genes (*GhMC02*, *GhMC04*, *GhMC07*, and *GhMC08*) were predominantly expressed in cotton fibers at 5–30 days post-anthesis (DPA). During environmentally induced PCD, the expression levels of four genes were affected in the root, stem, and leaf tissues within 6 h of an abiotic stress treatment. In general, the MC gene family affects the development of cotton fibers, including fiber elongation and fiber thickening while four prominent fiber- expressed genes were identified. The effects of the abiotic stress and hormone treatments imply that the cotton *MC* gene family may be important for fiber development. The data presented herein may form the foundation for future investigations of the *MC* gene family in *Gossypium* species.

## 1. Introduction

Programmed cell death (PCD) is an important factor affecting developmental biology and the maintenance of the steady state in continuously renewing tissues [1]. It is also essential for plant growth and development, survival following a pathogen invasion, and resistance against environmental stresses. There are two broad PCD categories in plants, namely developmentally regulated PCD (dPCD) and environmentally induced PCD (ePCD) [2]. A previous study revealed that the transcriptional signatures of dPCD are largely distinct from those of ePCD [3]. Despite the importance of PCD in plants, the underlying molecular mechanisms remain largely uncharacterized [4]. In contrast, the molecular mechanisms regulating PCD in animals have been elucidated based on analyses of the model system *Caenorhabditis elegans* [5]. In animal cells, caspases (aspartate-specific cysteine proteases) play a central role in PCD, including in the related signal transduction [6]. However, in plants, there have been few published studies regarding caspase genes, with the metacaspase (*MC*) gene family the only known plant caspase-like gene family [7]. Although MCs are similar to caspases regarding amino acid sequences and protein structures, they cannot be defined as caspases because they lack aspartate-specific proteolytic activity [8]. The *MC* gene family reportedly can be divided into two subfamilies [8], namely type I and type II, based on their sequences and structural similarities, with *AtMC1–3* representing type-I *MC* genes, and *AtMC4–9* categorized as type-II *MC* genes [8,9]. Both types of MCs have a putative conserved caspase-like catalytic domain composed of p10 and p20 subunits, which contain conserved catalytic histidine/cysteine residues [9,10]. Members of the *MC* gene family have been identified in a few model plant species and important crops. For example, in rice, eight *OsMC* genes were identified based on protein sequence alignments and were subsequently divided into two subtypes. An analysis of gene expression in different tissues revealed *OsMC* genes are specifically expressed in mature rice tissues, implying they may influence aging. Moreover, the expression levels of four *OsMC* genes are apparently upregulated by infections by *Magnaporthe oryzae* and insect pests as well as by drought and cold conditions, suggesting that these *OsMC* genes may contribute to plant defense responses [11,12]. In cucumber, five *CsMC* genes were identified according to aligned protein sequences, after which they were divided into two subtypes, type I (four or seven introns) and type II (one intron). The *CsMC* expression levels vary depending on the tissue as well as the developmental stage. An earlier investigation involving an abiotic stress treatment indicated that *CsMC* genes are involved in plant stress responses [13]. *Arabidopsis thaliana* consists of nine *MC* genes, of which *AtMC2* and *AtMC5* are specifically expressed in the roots and flowers; *AtMC1*, *AtMC3*, and *AtMC4* are expressed in all tested tissues; *AtMC8* is expressed at low levels in all tissues; *AtMC6* and *AtMC7* are highly expressed in the roots; and *AtMC9* is abundantly expressed in young leaves [14]. Of the six grape *MC* genes (*VvMC1–6*), *VvMC1,* and *VvMC2* are expressed in all examined tissues, whereas *VvMC3–6* exhibits tissue-specific expression [15]. In maize, MC-mediated proteolysis is a crucial part of leaf responses to both O_3_ and age-mediated senescence [16]. In tomato, researchers cloned a type-II *MC* gene, *LeMCA1*, from cDNA prepared from tomato leaves and suspension-cultured cells. An examination of genetic material from tomato leaves revealed that *LeMCA1* and several genes linked with PCD are located close together in suspension-cultured tomato cells, *LeMCA1* expression is not upregulated during chemical-induced PCD [17]. In pepper, the amino acid sequence of *CaMC9*, which is a positive regulator of pathogen-induced cell death, is 54% identical to *AtMC9*, and 40% consistent with plant type-II MCs [18]. In wheat, the expression of a type-II *MC* gene, *TaMCA4*, which was cloned from ‘Suwon11’ wheat, reportedly induces PCD via the expression of the mouse *bax* gene or a candidate *Puccinia striiformis* f. sp. *tritici* effector gene. Moreover, *TaMCA4* expression is upregulated in wheat leaves [19]. To date, there have been no published reports describing research on the cotton *MC* gene family.

Cotton is an important economic crop worldwide. We previously detected single nucleotide polymorphisms (SNP) in several candidate *MC* genes identified in stable quantitative trait loci, including *GhMC03* (*Gh_A06G1522*) and *GhMC20* (*Gh_A06G1524*) in *qFS-chr6-8*, implying the *MC* gene family may influence fiber strength [20]. In the current study, we systematically and comprehensively analyzed the *MC* family of *Gossypium* species regarding several factors, including chromosomal location, structure, and phylogeny. We also focused on the expression of these genes during responses to abiotic stress and fiber development.

## 2. Methods

### 2.1. Identification of Cotton MC Family Members

To identify homologs of *A. thaliana MC* genes [6], we applied the Hidden Markov Model of the HMMER 3.0 program [21] to search for the caspase-like domain (PF00656) in the following species: *Carica papaya* L. [22] (http://www.phytozome.net), *Theobroma cacao* [23], *Gossypium raimondii* [24] (http://www.phytozome.net), *Oryza sativa* [25] (http://www.mbkbase.org/R498/), *A. thaliana* [26] (http://www.arabidopsis.org), *Gossypium barbadense* [27] (http://database.chgc.sh.cn/cotton/index.html), *Gossypium hirsutum* [28,29] (http://mascotton.njau.edu.cn), and *Gossypium arboreum* [30] (ftp://bioinfo.ayit.edu.cn/downloads/) (Appendix A). The BLASTp program was used to verify sequences, with *AtMC* sequences as queries (e-value of 1e-5), after which incomplete sequences were manually removed. The remaining sequences were included in a multiple sequence alignment with the ClustalW program [31].

### 2.2. Phylogenetic Tree Construction, Gene Structure Analysis, and Conserved Motif Prediction

A multiple sequence alignment with the MEGA (version 6.06) program (http://www.megasoftware.net) [32] was used to construct a phylogenetic tree according to the neighbor-joining method, with 1000 bootstrap replicates. To examine the *MC* gene structures in *Gossypium* species, information regarding the exons and introns of the *MC* gene family members was retrieved from GFF3 files, and exon/intron structures were visualized with the Gene Structure Display Server 2.0 [33]. Conserved domains were predicted with the conserved domain database (http://www.ncbi.nlm.nih.gov/Structure/cdd/wrpsb.cgi) [34]. Motifs were analyzed with the MEME program (http://meme.nbcr.net/meme) [35], with the following parameters: maximum number of motifs, 20; minimum motif width, 6; and maximum motif width, 50.

### 2.3. Chromosomal Localization and Promoter Region Analysis

The chromosomal distribution of *MC* genes, which was determined based on genomic annotations, was visualized with the MapChart 2.2 program [36]. The 2,000-bp sequences upstream of the start codon of *MC* genes were extracted from the *G. hirsutum* genome sequences. The *cis*-acting elements were analyzed with the PlantCARE database (http://bioinformatics.psb.ugent.be/webtools/plantcare/html) [37].

### 2.4. Analysis of Repetitive Elements in MC Genes

To determine whether the *MC* gene family expanded through segmental duplication or tandem duplication events, a collinear analysis was completed with an all-to-all BLAST array (e-value of 1e-5) in the MCScan program [38].

### 2.5. Plant Materials and Treatments

To examine whether *MC* genes are differentially expressed in the fibers that vary in strength from two recombinant inbred lines, upland cotton lines 69307 (high fiber strength) and 69362 (relatively low fiber strength) [39,40] were grown under standard field conditions in Anyang, China. The flowering date was recorded as 0 days post-anthesis (DPA), and cotton bolls were collected in the morning every 5 days from 0 to 30 DPA. The fibers were isolated from bolls with a sterile knife. To investigate whether *MC* gene expression is affected by abiotic stresses, *G. hirsutum* cv. ‘0-153′, which is one of the parents of cotton line 69307, was grown in Hoagland nutrient solution [41] for 2 weeks before being exposed to simulated drought (20% PEG 6000) [42] or high salt (200 µM NaCl) [43] conditions or treated with the following phytohormones: 100 µM ABA, 100 µM naphthylacetic acid (NAA) [44], 100 µM salicylic acid (SA) [45], and 100 µM methyl jasmonate (MeJA) [46]. Three biological replicates of root, leaf, and stem tissues were collected at 0, 1, 6, 24, and 48 h after treatments. All samples were frozen in liquid nitrogen and stored at −80 °C.

### 2.6. Transcriptome Analyses and Quantitative Real-Time PCR

Transcriptome data were downloaded from the Sequence Read Archive (SRA) of the NCBI database (https://www.ncbi.nlm.nih.gov/) [47]. The SRA data for *G. barbadense* (PRJNA251673) [27], *G. hirsutum* (PRJNA248163) [29], *G. arboreum* (PRJNA179447) [30], and *G. raimondii* (PRJNA79005) [24] were converted to fastq data with the SRA Toolkit, after which the fastq reads were analyzed and filtered with the FASTX-Toolkit (http://hannonlab.cshl.edu/fastx_toolkit/index.html). The TopHat2 program [48] was used to map the clean data to the index genomes that were built with the Bowtie2 program [49], with the library-type and fr-unstranded parameters. The expression levels of the annotated genes in the reference genome were then calculated with the Cufflinks program [50]. To analyze the *MC* expression patterns in *Gossypium* species and determine whether they are tissue specific, the gene expression levels were visualized with the homogenized method involving log_2_(FPKM + 1) in the pheatmap program (https://CRAN.R-project.org/package=pheatmap). We designed *GhMC* gene-specific primers for a quantitative real-time polymerase chain reaction (qRT- PCR) assay (Appendix A), which was completed with the LightCycler® 480 II Real-time PCR instrument (Roche, Basel, Switzerland). Gene expression levels were calculated according to the 2^−ΔΔCt^ method, with three biological replicates and three independent PCR amplifications [51]. Significant differences (*p* <0.05) between the treated and untreated samples were determined with Student’s *t*-test in the SPSS 22.0 program. We set the 0 h time-point for the abiotic stress analysis and the line 69,307 fiber at 5 DPA as the controls.

## 3. Results

### 3.1. Identification of Cotton MC Genes

A bioinformatics analysis involving the Pfam database revealed that all deduced proteins had a caspase-like domain (PF00656), implying the corresponding genes belong to the *MC* family [52]. To characterize the *MC* genes in *Gossypium* species, 122 *MC* genes from eight species were identified (Appendix A), including 89 genes from the following cotton species: *G. hirsutum* (26 genes), *G. barbadense* (30 genes), *G. arboreum* (17 genes), and *G. raimondii* (16 genes). The *MC* coding regions were 633–2028 bp long, and the encoded proteins comprised 210–675 amino acids. Moreover, 61 *MC* genes were <3000 bp long, 21 *MC* genes were 3000–7000 bp long, and 7 *MC* genes exceeded 7000 bp (Appendix A).

### 3.2. Classification and Phylogenetic Analysis of the Cotton MC Gene Family

On the basis of phylogenetic relationships in the *MC* gene family, we confirmed the evolutionary relationships among eight species with a published sequenced genome. A multiple sequence alignment of 122 *MC* genes revealed two types of genes in these species (Appendix A, Figure 1a). Of the *MC* genes in the four analyzed cotton species, 42 are type-I *MC* genes, while the others are type-II *MC* genes (Figure 1b). Additionally, we detected 34, 2, 7, 6, and 41 homologs of *AtMC1*, *AtMC2*, *AtMC3*, *AtMC4*, and *AtMC9*, respectively, including 9, 1, 2, 2, and 12 *G. hirsutum* genes, respectively (Appendix A).

### 3.3. Conserved Motifs and MC Gene Structures

To clarify the structural diversity of the cotton *MC* genes, we analyzed the domains of the encoded MC protein sequences (Figure 2). On the basis of the predicted structures and relationships with *A. thaliana* genes, two types of *MC* genes were identified. There were 42 type-I *MC* genes and 47 type-II *MC* genes. Additionally, we determined that all of the type-I and type-II *MC* genes contain a sequence encoding the peptidase C14 domain. We also used the MEME program to identify the conserved motifs in the MC proteins from the four analyzed cotton species. A total of 20 motifs were detected (Figure 2c), motif 1, motif 3, and motif 5 were conserved motifs in all the MC genes in cotton, and motif 10 only in type-II MCs, yet motif 11, motif 14, motif 17 were merely in type-I MCs, with the type-I MCs carrying an N-terminal domain containing a proline-rich repeat motif as well as a zinc finger motif. In contrast, the type-II MCs lack such a domain, but harbor a linker region between the putative large (p20) and small (p10) subunits (Appendix A).

### 3.4. Analysis of Collinearity and Repetitive Elements in MC Genes

The evolutionary relationships between the *MC* gene families of tetraploid and diploid cotton species were assessed based on the similarity between family members, coverage, and chromosomal distances, which resulted in the identification of tandem repeats. The genes in the same chromosomal block (with an e-value <1e-5) were considered to be derived from tandem duplications, whereas those on different chromosomes resulted from segmental duplications. According to our MCScan analysis, *GaMC01* and *GaMC13* as well as *GaMC16* and *GaMC17* were tandem repeats in *G. arboreum*, and were located on chromosomes 06 and 03, respectively. In *G. hirsutum*, *GhMC09* and *GhMC18* were located on Dt chromosome 06. Additionally, in *G. raimondii*, *GrMC04* and *GrMC06*, *GrMC09,* and *GrMC12*, as well as *GrMC11* and *GrMC16* resulted from tandem duplication events, and were located on chromosome 10. Furthermore, *GaMC02*, *GrMC04*, *GhMC02*, and *GhMC04* were identified as orthologous genes, as were *GaMC05*, *GrMC05*, *GhMC07*, and *GhMC08* (Figure 3).

### 3.5. Analysis of MC Expression Patterns

To characterize the cotton *MC* expression patterns, transcriptome data were downloaded and re-analyzed. Specifically, an analysis of the *MC* genes expressed during the fiber development period in diploid cotton (0–15 DPA) and tetraploid cotton (10–28 DPA) (Figure 4a–d) revealed 3 of 17 *G. arboreum* genes, 3 of 16 *G. raimondii,* 3 of 30 *G. barbadense* genes, and 5 of 26 *G. hirsutum* genes, with standardized FPKM values >2 (Appendix A). Moreover, the homologous genes *GhMC02* and *GhMC04* as well as *GhMC07* and *GhMC08* were prominently expressed in the tetraploid cotton fiber(Appendix A) The average FPKM values of the four genes are the highest according to sheet 5 in Appendix A and combined with the data in sheet4 of Appendix A, the FPKM values of the four genes are expressed in the fiber expression of 10–28 DPA (Figure 4d), which are all greater than 10. It can thus be considered as a dominant expression gene, and *GhMC19* was specifically expressed during the early fiber development stage (Figure 4d,e).

On the basis of the above results, we speculated that some *GhMC* genes may be important for the development of cotton fibers. As we known, the development of cotton fiber is −3–50 days after anthesis, including fiber development initiation, fiber elongation, secondary wall thickening, and maturity period, of which 5–30 DPA is a critical period for fiber development and fiber quality formation [53].

We subsequently completed a qRT-PCR assay to verify the expression of all *GhMC* genes in two upland cotton recombinant inbred lines, namely 69307 (high fiber strength) and 69362 (relatively low fiber strength). We observed that 75% and 83.3% of the type-I *MC* genes tended to exhibit upregulated expression in the late period (20–25 DPA) in lines 69,307 and 69362, respectively (Appendix A). In contrast, the expression levels of 85.7% and 78.5% of the type-II *MC* genes were upregulated in the early stage (10–20 DPA) in lines 69307 and 69362, respectively (Appendix A). The 20 DPA time-point corresponds to the period in which cotton fibers elongate and the secondary wall thickens [52]. In the current study, our data suggested that the 20 DPA time-point represents the period in which both type-I and type-II *MC* genes are expressed in developing cotton fibers.

Two pairs of homologous genes (*GhMC02* and *GhMC04* as well as *GhMC07* and *GhMC08*) were determined to be predominantly expressed in cotton fibers based on transcriptome data and a transcriptional analysis during fiber development. Additionally, *GhMC02* and *GhMC04* are type-II genes that were most highly expressed at 10 DPA in lines 69307 and 69362, whereas *GhMC07* and *GhMC08* are type-I genes whose expression levels were highest at 20 and 10 DPA, respectively, in line 69307, and at 5 and 20 DPA, respectively, in line 69362 (Appendix A). Among the 26 *GhMC* genes, the following 13 genes were differentially expressed in the two analyzed lines: type-I genes: *GhMC06*, *GhMC07*, *GhMC08*, *GhMC21*, *GhMC22*, *GhMC24*, and *GhMC25*; type-II genes: *GhMC01*, *GhMC03*, *GhMC15*, *GhMC16*, *GhMC18*, and *GhMC20*.

We observed that *GhMC07*, *GhMC14*, and *GhMC22*, which are type-I genes, were more highly expressed in line 69307 (high fiber strength). Meanwhile, *GhMC05*, *GhMC13*, and *GhMC21* (type-I genes) as well as *GhMC01*, *GhMC02*, *GhMC04*, *GhMC09*, *GhMC10*, *GhMC15*, *GhMC16*, and *GhMC18* (type-II genes) were expressed at lower levels in line 69307. During the fiber development stage in line 69307, *GhMC25* and *GhMC03* were expressed at relatively low levels in the early stage but exhibited upregulated expression in the later stage. The *GhMC08* expression level was high in the early stage but was subsequently downregulated in the later stage. Furthermore, *GhMC12* expression levels varied considerably between 20 and 25 DPA.

The strengthening of cotton fibers involves a complex process [53], and the period in which fibers elongate and the secondary wall thickens (around 20 DPA) may be important for this process. The various types of *MC* genes may be differentially expressed in the cotton fibers of different species. Future studies will need to comprehensively evaluate this possibility.

### 3.6. Cis-Element Analysis in the Promoter Regions of MC Genes

On the basis of the difference between the transcriptome data and qRT-PCR results, we analyzed the *cis*-elements in the promoter regions of the above two types of *GhMC* genes (Figure 5). The following three types of *cis*-elements were identified: inducible, constitutive, and tissue-specific [54]. Specifically, the ABRE element (ACGTG) affects the responsiveness to abscisic acid (ABA) [55]. The LTR (long terminal repeat) element (CCGAAA) influences the responsiveness to low temperatures. The ARE (antioxidant response element) (AAACCA) is related to the anaerobic environment, whereas the TCA element (CCATCTTTTT) is related to the responsiveness to salicylic acid (SA) [56]. Meanwhile, the TGA-box (AACGAC) is an auxin-responsive element [57]. The two types of *GhMC* genes differed regarding the presence of a CGTCA-motif or TGACG-motif in the promoter, both of these elements contribute to the responsiveness to MeJA [58]. Furthermore, the TATC-box (TATCCCA) is associated with the responsiveness to gibberellin (GA) [59]. We analyzed the four genes predominantly expressed in fibers (*GhMC02*, *GhMC04*, *GhMC07*, and *GhMC08*). The *cis*-element associated with induction was responsive to GA, SA, and abscisic acid (ABA) for *GhMC02*, and SA and GA for *GhMC04*. Additionally, the *cis*-element related to drought was responsive to MeJA and ABA for *GhMC07* and *GhMC08*, and there were SA response-related components in *GhMC08*, but not in *GhMC07.* These observations indicated the analyzed genes vary regarding their *cis*-acting elements, even among the homologous genes.

### 3.7. Expression Analysis of Prominent Fiber-Expressed Genes under Abiotic Stress Conditions

Previous studies confirmed that *MC* genes are important regulators of PCD during stress responses in plants [9,15]. Moreover, PCD influences the aging process at all stages in plants. The *GhMC* expression levels in root, stem, and leaf tissues under abiotic stress conditions, including those induced by NaCl and PEG treatments, were analyzed by qRT-PCR (Figure 6). In response to the PEG treatment, the *GhMC* expression level rapidly increased, and peaked at 1 and 6 h, for all four examined *MC* genes in the root, stem, and leaf tissues. Similar expression trends were observed following the NaCl treatment (Figure 6b), with the highest expression levels in the root and stem tissues detected at 6 h. Drought stress induction is reportedly related to the ABA response pathway [60]. The *GhMC02*, *GhMC07*, and *GhMC08* promoters all include an ABA-responsive *cis*-element, while the *GhMC08* promoter also has an element related to drought (Figure 5). These results suggested that the two pairs of homologous genes analyzed in this study are responsive in various tissues soon after an exposure to abiotic stress.

### 3.8. Expression Analysis of Prominent Fiber-Expressed Genes Following Phytohormone Treatments

For a more in-depth study of *GhMC* expression levels induced by abiotic stress, the expression patterns of four *GhMC* genes after ABA, naphthylacetic acid (NAA), SA, and MeJA treatments were analyzed by qRT-PCR. Published research suggests that type-II MCs are not directly responsible for the earlier reported caspase-like activities in plants, but they are important for ePCD [61]. On the basis of the analyses of the promoter elements (Figure 5), we examined the effects of various hormones on the expression of *GhMC* genes. We observed that the relative expression levels for the four genes were rapidly upregulated 1 h after the ABA treatment and stayed at that level until the 48-h time-point. However, these genes were differentially responsive to ABA, and the responses in the leaves were greater than those in the stems and roots. The NAA treatment upregulated the relative *GhMC02*, *GhMC04*, and *GhMC08* expression levels in the roots in 24 h (Figure 7b), and *GhMC07* and *GhMC08* were relatively highly expressed in the stem and leaf tissues, respectively. In response to SA, the peak relative expression levels of the four examined genes occurred at 1 h in the stems. However, the expression of the four genes responded differently in the root tissue, with peak *GhMC02*, *GhMC04*, and *GhMC07* transcript levels detected at 24 h. Regarding the leaves, the highest expression levels of the homologous genes *GhMC02* and *GhMC04* were observed 24 h after the hormone treatment, whereas the peak expression levels of the other pair of homologous genes occurred at 1 h (Figure 7c). For the Me-JA treatment, the expression levels of the four genes increased rapidly at 6 h in the roots. The maximum relative expression levels in the stem tissue were observed at 6 h for *GhMC02* and *GhMC04*, and at 1 h for *GhMC07* and *GhMC08*. In leaves, *GhMC02*, *GhMC04*, and *GhMC08* were relatively highly expressed at 6 h (Figure 7d). These results implied that following various hormone treatments, the analyzed genes had diverse response times in the root, stem, and leaf tissues.

## 4. Discussion

### 4.1. Number of Cotton MC Genes

Metacaspases are cysteine proteases that are widely distributed among Viridiplantae species, from algae to vascular plants [62]. However, there have been no systematic analyses of cotton MCs. In this study, we completed a genome-wide identification and expression analysis of the cotton *MC* gene family. We identified 89 *MC* genes in the following four cotton species: *G. hirsutum* (26 genes), *G. barbadense* (30 genes), *G. arboreum* (17 genes), and *G. raimondii* (16 genes). Additionally, 42 and 47 of these genes were type-I and type-II *MC* genes, respectively. A comparison of the *MC* genes from 42 plant species revealed there are more than twice as many type-I *MC* genes than type-II *MC* genes, indicating that most plant species have more type-I MCs than type-II MCs [9]. However, in this study, the numbers of type-I and type-II MCs were similar, possibly because only *AtMC4* and *AtMC9* homologs were detected as type-II *MC* genes in the four examined cotton species.

Previous studies determined that *MC* genes are differentially expressed in various tissues [62,63]. In grape, type-I *MC* genes are highly expressed in the stem [14]. In rubber trees, *HbMC8* is highly expressed in all examined tissues, while *HbMC5* and *HbMC6* are most highly expressed in the leaves [62]. In rice, eight *OsMC* genes, which were identified based on a protein sequence alignment, were subsequently divided into two subtypes. An analysis of the expression profiles of these eight genes in various tissues revealed specific expression patterns in mature rice tissues [11,12]. Additionally, five *CsMC* genes were identified and classified into two subtypes, with 4–7 introns in the type-I *CsMC* genes, and only one intron in the type-II *CsMC* genes [13].

### 4.2. Metacaspase Genes and Fiber Development in *Gossypium hirsutum L.*

The data presented herein indicate that different types of *MC* genes may have variable expression patterns in diverse cotton fibers. We previously determined single nucleotide polymorphism locations on a genetic map, thereby identifying stable quantitative trait loci for cotton fiber quality [64,65]. Moreover, some candidate genes reportedly belong to the *MC* gene family, including *GhMC03* and *GhMC20* in *qFS-chr6-8* [20]. These observations suggest that the *MC* gene family may have important effects on fiber strength traits. On the basis of the process underlying cotton fiber cell development, we speculated that *MC* genes are differentially expressed during fiber development, specifically during the strengthening of the fibers. Cotton fiber develops from a single cell, with stages corresponding to initiation, elongation, thickening, and maturity; these stages are associated with cell development, maturation, and senescence [53]. Additionally, the *MC* genes are reportedly important for cell senescence and apoptosis [66]. Thus, we speculated that the strengthening of cotton fibers is also regulated by this process. Through the data analysis of the transcriptome, the level of the MC gene family does not seem to be highly expressed, but by analyzing the expression level of two RIL (Recombinant Inbred Lines) materials (69307 and 69362) screened for many years, we found some genes with an up-regulation trend during fiber development (5-30DPA). For example, the expression level of *GhMC05*, *GhMC06 GhMC23* are up-regulated on 20-25DPA, *GhMC21* is up-regulated on 30 DPA for 69362. These results provide a variety of possibilities for subsequent multi-angle studies of the MC gene.

Fiber strength is the key factor influencing cotton fiber quality, and the key fiber development period during which fiber cells lengthen and thicken occurs between 10 and 25 DPA [53]. Regarding the two types of *GhMC* genes, more than 90% of these genes are maximally expressed at 10–25 DPA. Moreover, the type-II *MC* genes are crucial for fiber elongation, whereas the type-I *MC* genes contribute to secondary wall thickening. Our transcriptome analysis indicated the FPKM of *GhMC02*, *GhMC04*, *GhMC07*, and *GhMC08* exceeded 10 (Appendix A). Furthermore, *GhMC07* and *GhMC08*, which are type-I genes, are homologs of *AtMC1*. During dPCD, *AtMC1* is expressed in parallel pathways, both positively regulating the hypersensitive response (HR)-mediated cell death in young plants when the functions are not masked by the cumulative stresses of aging, and negatively regulating senescence in older plants [67]. The *GhMC02* and *GhMC04* type-II genes are homologs of *AtMC4*, which encodes a positive regulator of biotic and abiotic stress-induced PCD, similar to animal caspases [68]. Because *AtMC4* appears to be ubiquitously expressed in all organs, the *AtMC4* protein might also regulate many other biological processes in *A. thaliana* [69]. Therefore, our results imply that the *MC* gene family may have crucial effects on cotton fiber quality.

### 4.3. Effects of Abiotic Stresses and Hormones on the Metacaspase Gene Family

Because of genetic redundancy, PCD is rarely observed in *A. thaliana MC* knockout lines. However, *AtMC1* and *AtMC2* can enhance as well as inhibit PCD, with *AtMC1* positively regulating cell death, and *AtMC2* exhibiting the opposite effect [70]. Moreover, the expression of *AtMC4* reportedly accelerates cell death in response to biotic and abiotic stresses [71]. In cucumber, the peak *CsMC1* expression level occurs at 3 h after a NaCl treatment and is maintained until 12 h. In contrast, *CsMC2–4* expression is gradually downregulated until 6 h, and then slightly upregulated at 12 h. Following a PEG treatment, the *CsMC1–5* expression levels are similar to those induced by NaCl [13].

Plant hormones affect all stages of the aging process. Additionally, exogenous ABA can promote leaf detachment. During leaf senescence, the expression levels of a 9-*cis*-epoxycarotenoid dioxygenase gene (*NECD*) and two ABA aldehyde oxidase genes (*AAO1* and *AAO3*) are upregulated. These enzymes are critical for ABA synthesis. Meanwhile, the expression of the gene encoding an ABA-induced receptor kinase (*RPK1*) is upregulated, and the resulting enzyme accelerates leaf senescence, as does ABA-induced H_2_O_2_ [72]. Regarding *GhMC02*, in response to an ABA treatment, the peak expression level occurred at 1 h in leaves, while the expression levels in the root and stem tissues remained upregulated until 48 h. These results suggest that the ABA response occurs after other types of induction in the roots and stem, although this possibility will need to be experimentally verified. Therefore, the *cis*-acting element differed among the analyzed genes, even between homologous genes. In *A. thaliana*, increases in the jasmonate (JA) content activate JA-biosynthesis enzymes to varying extents and induce the expression of *SAG* genes related to senescence, ultimately resulting in leaves exhibiting the characteristics of aging [73]. Our analysis of promoter elements indicated there are no MeJA-responsive elements in the promoters of *GhMC02* and *GhMC04*, but the response time of these genes in various tissues was 6 h following a MeJA treatment, implying that a regulatory pathway induced by an unknown mechanism is involved. Salicylic acid affects the hypersensitive response PCD, and cell death occurring during plant disease-resistant HR is a type of PCD [74]. Additionally, in *A. thaliana*, the SA concentration is 4-times higher in senescing leaves than in non-senescing leaves, suggesting that this hormone contributes to aging and cell death. The mechanism underlying SA-regulated leaf senescence is very similar to that mediating the natural aging process of leaves [75,76]. The *GhMC07* promoter lacks an SA-responsive element, but the application of exogenous SA induced the expression of this gene in 1 h in the stem and leaf tissues, but in 24 h in the roots. Increased auxin (IAA) levels due to the upregulated expression of genes in the IAA biosynthesis pathway, such as *TSA1* (tryptophan synthase), *AO1* (indoleacetate oxidase), and *NIT1–3* (acetonitrile hydrolases), led to delayed leaf senescence [77]. In the current study, auxin- associated *cis*-elements were not detected in *GhMC02*, *GhMC04*, *GhMC07*, and *GhMC08*. Nevertheless, following an NAA treatment, the response time was 24 h and 1 h in the root and stem tissues, respectively. Moreover, *GhMC04* expression in leaves was unaffected by NAA.

## 5. Conclusions

In this study, we comprehensively characterized the cotton *MC* genes by analyzing their phylogenetic relationships, conserved motifs, gene structures, promoter sequences, and expression profiles in cotton lines with varying fiber strength. We also assessed the effects of abiotic stress and hormone treatments. This study represents a thorough bioinformatics- and gene expression-based investigation of the *MC* gene family. The data presented herein provide the foundation for future studies on gene functions during cotton fiber development.

## Figures and Tables

**Figure 1 genes-10-00527-f001:**
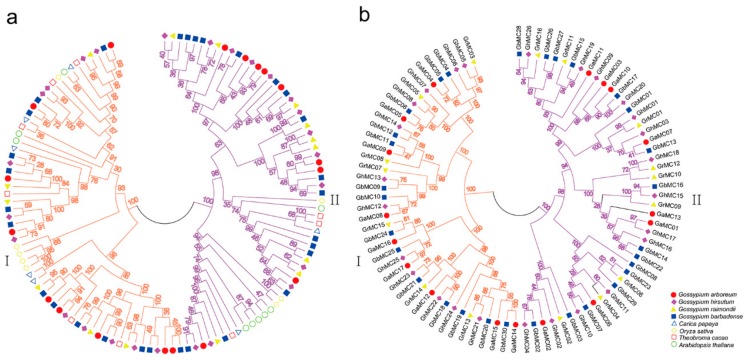
Phylogenetic tree of the metacaspase gene family. (**a**) Neighbor-joining phylogenetic tree of metacaspase genes from eight species. (**b**) Neighbor-joining phylogenetic tree of the cotton metacaspase gene family.

**Figure 2 genes-10-00527-f002:**
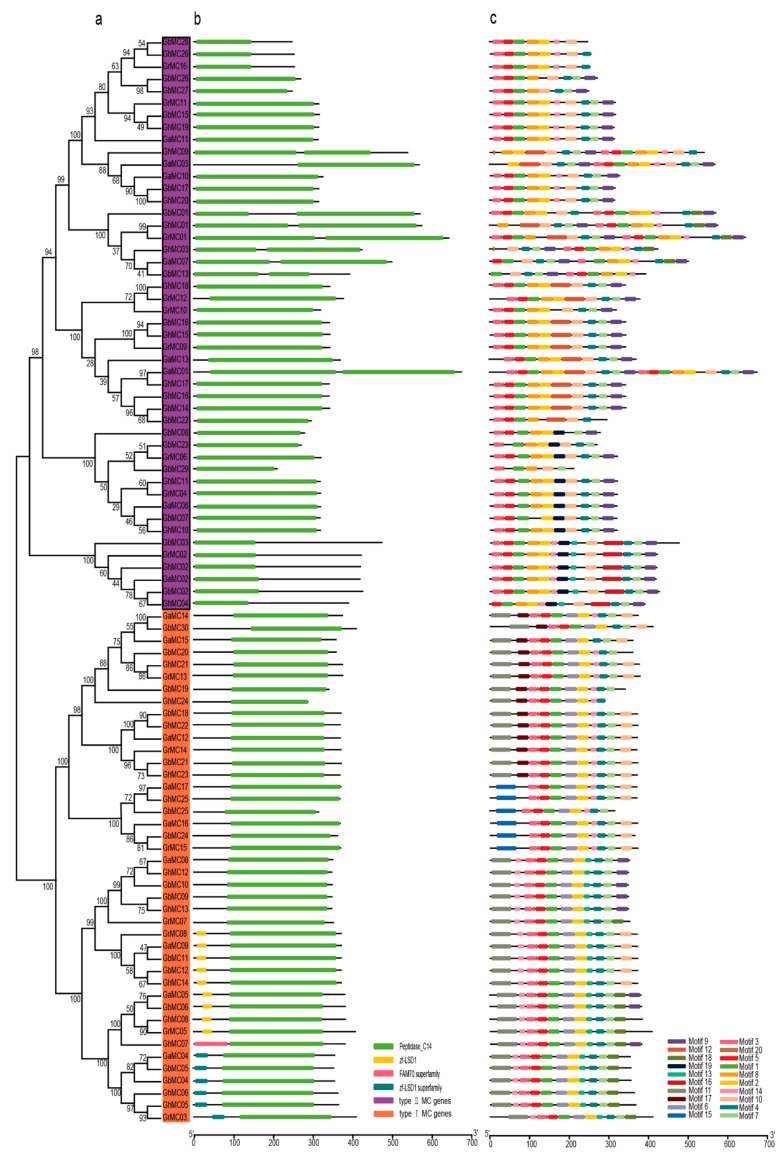
The stucture analysis of metacaspase gene family. (**a**) Phylogenetic tree of the metacaspase (MC) gene family. (**b**) Analysis of the protein sequences of the MC gene family domains. (**c**) Analysis of the protein sequences of the MC gene family motifs.

**Figure 3 genes-10-00527-f003:**
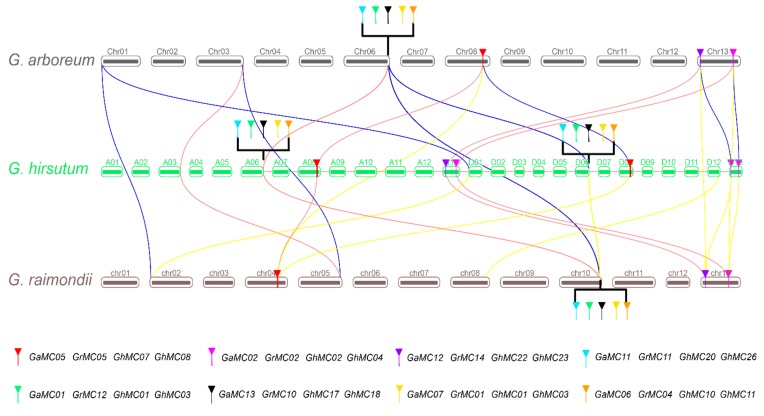
Collinearity of metacaspase genes in tetraploid and diploid cotton species.

**Figure 4 genes-10-00527-f004:**
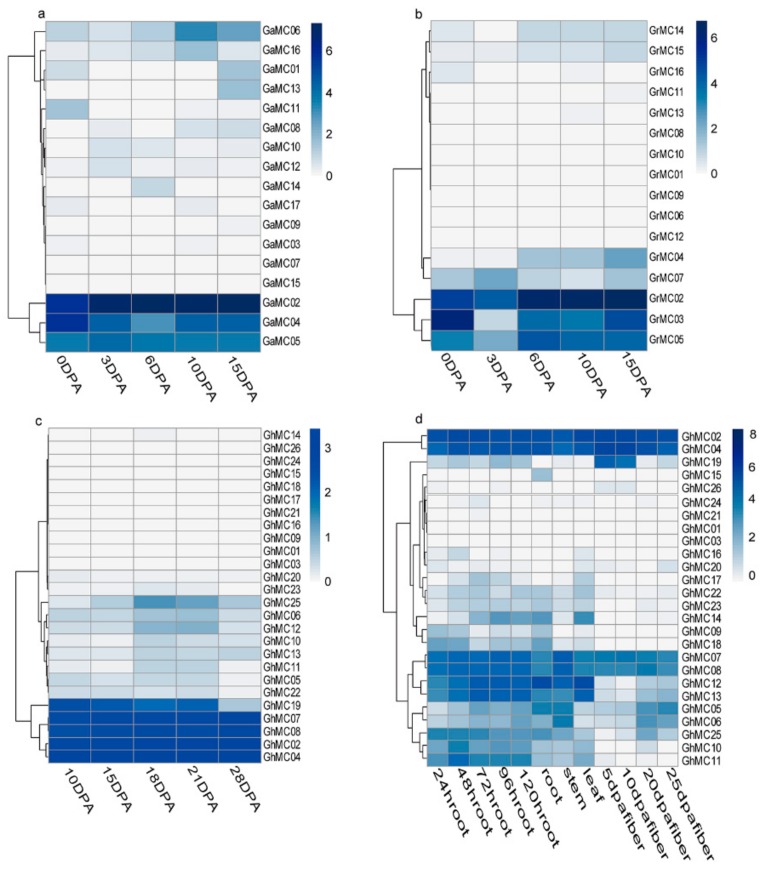
Heatmap presenting the metacaspase gene expression levels based on a transcriptome analysis of log_2_(FPKM+1). (**a**) Metacaspase gene expression levels during fiber development in *Gossypium arboreum* (0–15 DPA). (**b**) Metacaspase gene expression levels during fiber development in *Gossypium raimondii* (0–15 DPA). (**c**) Metacaspase gene expression levels during fiber development in *Gossypium barbadense* (10–28 DPA). (**d**) Metacaspase gene expression levels during fiber development in *Gossypium hirsutum* (10–28 DPA). (**e**) Metacaspase gene expression levels in the root, stem, leaf, and fiber tissues from *G. hirsutum*.

**Figure 5 genes-10-00527-f005:**
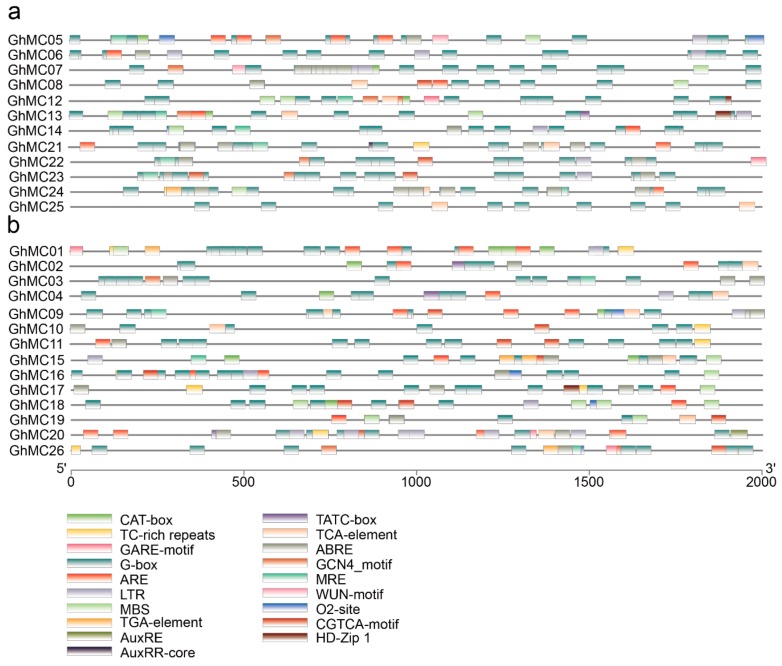
Analysis of cis-elements in *GhMC* genes. (**a**) Type-I *GhMC* genes; (**b**). Type-II *GhMC* genes.

**Figure 6 genes-10-00527-f006:**
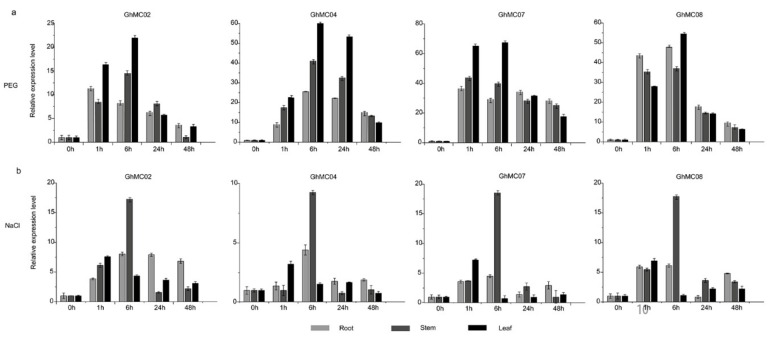
Analysis of *GhMC* expression levels following abiotic stress treatments according to a qRT- PCR assay. (**a**) 20% PEG 6000 treatment; (**b**) 200 µM NaCl treatment.

**Figure 7 genes-10-00527-f007:**
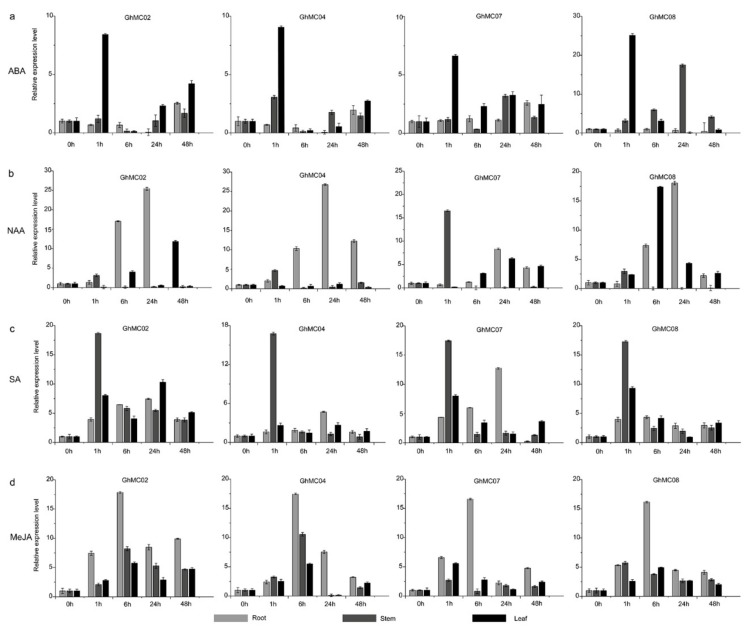
Analysis of GhMC expression levels following phytohormone treatments according to a qRT-PCR assay. (**a**) 100 µM ABA (abscisic acid) treatment; (**b**) 100 µM NAA (naphthylacetic acid) treatment; (**c**) 100 µM SA (salicylic acid) treatment; (**d**) 100 µM MeJA (methyl jasmonate) treatment.

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
