# Peer review of "Genome-Wide Identification and Expression Analysis of the Metacaspase Gene Family in Gossypium Species"

_genes, 2019, doi:10.3390/genes10070527_

Round 1

Reviewer 1 Report

In this manuscript, the authors comprehensively identified MC genes from four cotton species based on their genomic sequence similarity and database search. Following the discovery of the MC genes, they conducted phylogenetic analysis, protein domain/motif analysis, promoter analysis to characterize and compare the primary sequence differences between different MC genes across cotton species. They also checked the expression profiles of the MC genes at different stages of the cotton fiber development between different species and strains. At last, they assessed the effects of abiotic stress and hormone treatments on MC gene expression. This study provides a comprehensive knowledge about the MC gene family in cotton species and could potentially serve as a valuable resource for researchers in the field of cotton and plant biology. Below are my specific comments:

Fig. 2 is a little confusing. Based on the annotated domains and motifs of each gene, I assume the authors are showing the protein sequence alignment in Fig.2b and Fig. 2c, but the figure legend says “Analysis of the genomic sequences of the MC gene family”. Please clarify this. In Line 177 the authors say “the type-II MC genes mainly comprise a sequence encoding the peptidase C13 domain.” However I couldn’t find the peptidase C13 domain in the figure legends. Also it seems all MC genes consist of the peptidase C14 domain, not just the type-I MC genes.

 In Fig. 3, what do the different colors of lines stand for? It’s also quite difficult to tell which gene are the tandem repeats. Maybe a zoom-in of such regions will help? Clear annotation of the orthologous genes in different species is also helpful.

Please indicate what do the numbers in Fig. 4 standard for. Are they raw FPKM values or log(FPKM)? What are the rankings/percentile of the “prominently expressed ” MC genes in all cotton genes expressed at the same time? To me, all MC genes seem to be lowly expressed in Fig. 4.

In Fig. S2, many GhMC genes are upregulated during the fiber elongation, such as GhMC15, GhMC23, GhMC14 and GhMC10, regardless of the cotton strains. However they are not significantly upregulated in the RNA-seq data shown in Fig. 4, GhMC14 and GhMC15 are barely expressed. How will the authors explain such big differences?

Author Response

Response to Reviewer 1 Comments

Point 1: Fig. 2 is a little confusing. Based on the annotated domains and motifs of each gene, I assume the authors are showing the protein sequence alignment in Fig.2b and Fig. 2c, but the figure legend says “Analysis of the genomic sequences of the MC gene family”. Please clarify this. In Line 177 the authors say “the type-II MC genes mainly comprise a sequence encoding the peptidase C13 domain.” However I couldn’t find the peptidase C13 domain in the figure legends. Also it seems all MC genes consist of the peptidase C14 domain, not just the type-I MC genes.

Response 1We are very grateful to you for your valuable comments on our manuscript. For the legend of Figure 2, we change it to “Analysis of the protein sequences of the MC gene family” in line 190 and 191. About the peptidase C14 domain and peptidase C13 domain, we corrected it for “all of the type-I and type-II MC genes contain a sequence encoding the peptidase C14 domain” in line 18, 19, 178 to 180, respectively.

Point 2: In Fig. 3, what do the different colors of lines stand for? It’s also quite difficult to tell which gene are the tandem repeats. Maybe a zoom-in of such regions will help? Clear annotation of the orthologous genes in different species is also helpful.

Response 2: In Figure 3, the results of the tandem repeats cant be clearly expressed because of the different colors of lines stand for genes from different species. For this, we explained in the manuscript; and as for the result of the segment repeats, we have made appropriate modifications for Figure 3 in line 204, sharing 8 different color labels, so that visually see the evolutionary sources of orthologous genes.

Point 3: Please indicate what do the numbers in Fig. 4 standard for. Are they raw FPKM values or log (FPKM)? What are the rankings/percentile of the “prominently expressed” MC genes in all cotton genes expressed at the same time? To me, all MC genes seem to be lowly expressed in Fig. 4.

Response 3: In Figure 4, the number indicate the log 2 (FPKM+1) and the data shown in Table S2 is the original FPKM value, we express them in line 224 and Table S2, respectively. For the “prominently expressed”, the average FPKM values of the four genes are the highest according to the sheet 5 in Table S2 and combined with the data in sheet4 of Table S2, the FPKM values of the four genes are expressed in the fiber expression of 10-28 DPA , which are all greater than 10, so it can be considered as a dominant expression gene. We have made the appropriate changes in the manuscript in line 216 to 220.

Point 4: In Fig. S2, many GhMC genes are upregulated during the fiber elongation, such as GhMC15, GhMC23, GhMC14 and GhMC10, regardless of the cotton strains. However they are not significantly upregulated in the RNA-seq data shown in Fig. 4, GhMC14 and GhMC15 are barely expressed. How will the authors explain such big differences?

Response 4: The material used for RNA-seq analysis shown in Figure 4 is the standard line TM-1 of upland cotton, but the two materials we used have a significant difference in fiber strength from the RIL population which have been selected for many years, perhaps, their fiber developmental regulation process are very different from TM-1, so there may be difference between Q-pcr and RNA-seq data in the manuscript. Regardless of the cotton strains, RNA-seq analysis and qPCR are the two different detection methods, and it is normal and reasonable to have 30~40% inconsistency. (Su Z, Łabaj P P, Li S, et al. A comprehensive assessment of RNA-seq accuracy, reproducibility and information content by the Sequencing Quality Control Consortium[J]. Nature biotechnology, 2014, 32(9): 903.).

Thank you again!!

Reviewer 2 Report

The authors of this manuscript claim to have performed analysis of metacaspases in four cotton species. I have several conserns and I list them below:

I have not manually checked what proteins have been identified as having the C13 domain, but these are not metacaspases! Metacaspases are by definition (!) proteins containing the p20 domain and are thus classified as belonging to the C14 family in the MEROPS database.This should be revised as it may change the story of the whole manuscript.

When all the genes are analysed , e.g. the data represented in Fig. 4, the G. barbadense is missing without any explanation why.

In methods also papaya, cacao and rice are mentioned, but never presented. Why?

As a measure of stres authors use anthesis. For metacaspase involved community or for those who are not familier with cotton, more information shoud be given about this approach since very little information are given to the reader.

Figure 2 is generally welcome, but not informative enough. Letters should be better seen and the (c) part gives no information at all. Motif numbers are not good, neither do the supplementing figures give any more information. What are those motifs?

Authors often mention tetraploid and diploid cotton species. Why is this important?

Apperance of "double metacaspases" are more than intriguing and this is the first publication that identifies them in any species. More emphasis should be given to them and their expression levels (although they do not seem to be highly expressed during tested conditions).

Now it is well known that type I as well as type II metacaspases are sub-divided into different classes (check Klemencic and Funk, 2019, JXB); discussion should explain more how do AtMC4 and AtMC9 homologues in cotton resemble other known metacspases and their function.

The second sentence in abstract does not make sense (line 15).

Author Response

Response to Reviewer 2 Comments

Point1: I have not manually checked what proteins have been identified as having the C13 domain, but these are not metacaspases! Metacaspases are by definition (!) proteins containing the p20 domain and are thus classified as belonging to the C14 family in the MEROPS database.This should be revised as it may change the story of the whole manuscript.

Response 1: We sincerely thanks for your reviewing efforts and the approval of our manuscript. About the relationship between C13 domain and metacaspase, we have made relevant modifications in line 18 and 19, 178 to 180 in the manuscript.

Point 2: When all the genes are analysed, e.g. the data represented in Fig. 4, the G. barbadense is missing without any explanation why.

Response 2: We have changed Figure 4, the analysis of G. barbadense is shown in Figure 4c, and the original FPKM value of G. barbadense are added to sheet3 of Table S2, and the related legend changes in line 224 to 229.

Point 3: In methods also papaya, cacao and rice are mentioned, but never presented. Why?

Response 3: In order to obtain the protein sequences belong to the MC gene family to construct a phylogenetic tree (Figure 1a) in papaya, cacao and rice, we mentioned these species in the material section. We also add corresponding files (Supplementary file 1, Supplementary file 2) in the manuscript in lines 97 and 164. In introduction and discussion, we mentioned the research in rice, but we don, t find related research in papaya and cacao of MC gene gamily, so there are not mentioned in the manuscript.

Point 4: As a measure of stres authors use anthesis. For metacaspase involved community or for those who are not familier with cotton, more information shoud be given about this approach since very little information are given to the reader.

Response 4: We add instructions in the manuscript from line 231 to line 234. (As we known, the development of cotton fiber is -3-50 days after anthesis, including fiber development initiation, fiber elongation, secondary wall thickening, and maturity period, of which 5-30 DPA is a critical period for fiber development and fiber quality formation).

Point 5: Figure 2 is generally welcome, but not informative enough. Letters should be better seen and the (c) part gives no information at all. Motif numbers are not good, neither do the supplementing figures give any more information. What are those motifs?

Response 5: We changed the resolution of Figure 2 to make it look sharper. In addition, we added a description of the motif for Figure 2C in line182 in the manuscript.

Point 6: Authors often mention tetraploid and diploid cotton species. Why is this important?

Response 6: Cotton is an important economic crop, and Gossypium arboreum is an early cultivar in China and IndiaGossypium raimondii is a donor of the D subgenome in tetraploid cotton, which are the important materials for cotton biology research. Gossypium hirsutum with good agronomic traits and high yield, Gossypium barbadense is a provider of high-grade cotton materials, which are the widely cultivated cotton cultivar, so, we repeatedly emphasized that diploid and tetraploid cotton play a more important role in elucidating the evolutionary analysis of the MC gene family.

Point 7: Apperance of "double metacaspases" are more than intriguing and this is the first publication that identifies them in any species. More emphasis should be given to them and their expression levels (although they do not seem to be highly expressed during tested conditions).

Response 7: We add instructions in the manuscript from line 363 to line 369.

Point 8: Now it is well known that type I as well as type II metacaspases are sub-divided into different classes (check Klemencic and Funk, 2019, JXB); discussion should explain more how do AtMC4 and AtMC9 homologues in cotton resemble other known metacspases and their function.

Response 7: So far, we have not found a functional verification study of AtMC4 and AtMC9 homologues in cotton, the annotation information we obtained is also based on the homologous genes of Arabidopsis, so we will not repeat them in the discussion section.

Point 9: The second sentence in abstract does not make sense (line 15).

Response 9: We have deleted the sentence.

Thank you very much!